# Assessing the Feasibility and Efficacy of Pre-Sleep Dim Light Therapy for Adults with Insomnia: A Pilot Study

**DOI:** 10.3390/medicina60040632

**Published:** 2024-04-14

**Authors:** Jihyun Yoon, Seok-Jae Heo, Hyangkyu Lee, Eun-Gyeong Sul, Taehwa Han, Yu-Jin Kwon

**Affiliations:** 1Department of Family Medicine, Anam Hospital, Korea University College of Medicine, Seoul 02481, Republic of Korea; ghyun.yoon@kumc.or.kr; 2Division of Biostatistics, Department of Biomedical Systems Informatics, Yonsei University College of Medicine, Seoul 03722, Republic of Korea; sjheo@yuhs.ac; 3College of Nursing, Mo-Im Kim Research Institute, Yonsei University, Seoul 03722, Republic of Korea; hkyulee@yuhs.ac; 4Department of Family Medicine, Yongin Severance Hospital, Yonsei University College of Medicine, Yongin 16995, Republic of Korea; eksul1982@yuhs.ac; 5Integrative Research Center for Cerebrovascular and Cardiovascular Diseases, Yonsei University College of Medicine, Seoul 03722, Republic of Korea

**Keywords:** insomnia, light therapy, sleep quality, melatonin, clock gene expression

## Abstract

*Background*: Insomnia is increasingly recognized for its marked impact on public health and is often associated with various adverse health outcomes, including cardiovascular diseases and mental health disorders. The aim of this study was to investigate the efficacy of pre-sleep dim light therapy (LT) as a non-pharmacological intervention for insomnia in adults, assessing its influence on sleep parameters and circadian rhythms. *Methods*: A randomized, open-label, two-arm clinical trial was conducted over two weeks with 40 participants aged 20–60 years, all of whom had sleep disorders (CRIS, KCT0008501). They were allocated into control and LT groups. The LT group received exposure to warm-colored light, minimizing the blue spectrum, before bedtime. The study combined subjective evaluation via validated, sleep-related questionnaires, objective sleep assessments via actigraphy, and molecular analyses of circadian clock gene expression in peripheral blood mononuclear cells. Baseline characteristics between the two groups were compared using an independent *t*-test for continuous variables and the chi-squared test for categorical variables. Within-group differences were assessed using the paired *t*-test. Changes between groups were analyzed using linear regression, adjusting for each baseline value and body mass index. The patterns of changes in sleep parameters were calculated using a linear mixed model. *Results*: The LT group exhibited significant improvements in sleep quality (difference in difference [95% CI]; −2.00 [−3.58, −0.43], and sleep efficiency (LT: 84.98 vs. control: 82.11, *p* = 0.032), and an advanced Dim Light Melatonin Onset compared to the control group (approximately 30 min). Molecular analysis indicated a significant reduction in CRY1 gene expression after LT, suggesting an influence on circadian signals for sleep regulation. *Conclusions*: This study provides evidence for the efficacy of LT in improving sleep quality and circadian rhythm alignment in adults with insomnia. Despite limitations, such as a small sample size and short study duration, the results underscore the potential of LT as a viable non-pharmacological approach for insomnia. Future research should expand on these results with larger and more diverse cohorts followed over a longer period to validate and further elucidate the value of LT in sleep medicine. Trial registration: The trial was registered with the Clinical Research Information Service (KCT0008501).

## 1. Introduction

Insomnia, a pervasive sleep disorder characterized by persistent difficulties in initiating or maintaining sleep, poses a substantial public health challenge worldwide [1]. Beyond its immediate impact on sleep quality, chronic insomnia is associated with a range of adverse health outcomes, including cardiovascular diseases, mental health disorders, and impaired cognitive function [2,3].

Non-pharmacological interventions are gaining attention for their potential efficacy and fewer side effects compared to pharmacotherapy [4]. Light therapy (LT) [5], in which specific wavelengths of light are used to regulate circadian rhythms and influence sleep patterns, has emerged as a promising avenue in this regard [4,6]. The circadian system, governed by the master clock in the brain, orchestrates the intricate balance between wakefulness and sleep [7,8]. Moreover, the circadian timing system is known to play a crucial role in regulating lipid metabolism and energy homeostasis [9,10,11]. Disruptions to this system, often induced by the artificial lighting prevalent in contemporary life, have been implicated in the increasing prevalence of sleep disorders, including insomnia [6]. Sleep disorders affect 6% to 20% of populations worldwide, including in South Korea (one in five adults) [12,13,14]. This emphasizes the necessity for interventions replicating the benefits of natural light [8].

LT can be tailored in terms of intensity and correlated color temperature (CCT) for the management of sleep problems [4,15,16]. Research, including our own [4,15,16], has demonstrated that light with a wavelength of approximately 460 nm can significantly modulate melatonin suppression, thereby facilitating sleep regulation. The role of CCT in LT, influencing melatonin levels and circadian rhythms, is increasingly recognized as a crucial factor in sleep therapy [17]. However, research on low-intensity, warm-colored LT, especially at night, is not as extensive [18,19]. Studies suggest that nighttime lighting depleted of blue light can reduce nocturnal awakenings and minimize disruption to circadian and sleep parameters [18,20,21].

Given that dim lighting may offer a more practical and less disruptive treatment option for patients with sleep disorders [22], potentially improving adherence and outcomes, giving more attention to the subject is warranted [22].

The aim of this study was to investigate the efficacy of pre-sleep dim light therapy (LT) as a non-pharmacological intervention for insomnia in adults, evaluating its impact on sleep parameters and circadian rhythms. We also aim to investigate the impact of LT on metabolic parameters, hormone levels, and the expression of clock genes to comprehensively understand the role of LT in human health.

## 2. Materials and Methods

### 2.1. Study Population

This study constituted a randomized, open-label, two-arm clinical trial conducted over a two-week period. The study protocol was approved by the Institutional Review Board of Yongin Severance Hospital (IRB No. 9-2023-0039) and was registered with the Clinical Research Information Service (CRIS, KCT0008501). The study adhered to the principles of the Declaration of Helsinki, and all participants provided informed consent for participation before enrolment. Patient recruitment took place from 16 June 2023 to 18 September 2023 at Yongin Severance Hospital in Yongin, South Korea.

We enrolled participants who presented at a primary care clinic seeking medical assistance for persistent sleep issues lasting over three months, resulting in considerable distress or disruption to their daily activities. Our focus was on primary insomnia, a chronic sleep disorder distinguished by challenges in initiating or maintaining sleep, or experiencing non-restorative sleep, without identifiable medical and psychiatric diseases that could affect sleep. The patients were asked whether they had a medical history of diagnoses from a physician. We did not included sleep disorders specified in the 5th edition of the Diagnostic and Statistical Manual of Mental Disorders, except for primary insomnia disorder (e.g., hypersomnolence disorder, narcolepsy, obstructive sleep apnea/hypopnea, central sleep apnea, sleep-related hypoventilation), sleep disorders linked to other mental disorders, sleep disorders associated with other medical conditions, a history of psychiatric illness, and initiation of other treatments for insomnia within the preceding three months.

### 2.2. Clinical and Biochemical Analyses

Study visits were scheduled at screening, baseline, and after two weeks. Body weight and height were measured, and body mass index (BMI) was calculated. Systolic and diastolic blood pressure were measured with the participant in a seated position using the right arm after at least 5 min of rest. Lifestyle factors, including smoking, alcohol consumption, and exercise, as well as underlying diseases, such as diabetes, hypertension, and dyslipidemia, were assessed via a self-report questionnaire. Participants were categorized as current smokers or non-smokers and current drinkers or non-drinkers. Physical activity was defined as exercising for more than 30 min at a time and more than three times a week. The histories of hypertension, dyslipidemia, and diabetes were all used as binary variables.

Blood samples were collected after a fasting period of more than 8 h. White blood cell counts were quantified via flow cytometry using the XN2000 hematology analyzer (Sysmex, Kobe, Japan). Insulin concentrations were analyzed using chemiluminescent microparticle immunoassays and the Architect i2000SR immunoassay analyzer (Abbott, Abbott Park, IL, USA). Concentrations of lipids, including those of total cholesterol, low-density lipoprotein cholesterol, non-high-density lipoprotein cholesterol, triglycerides, and high-density lipoprotein cholesterol, were analyzed using an enzymatic color test. C-reactive protein concentrations were assessed using an immunoturbidimetric method. The homeostatic model assessment of insulin resistance was calculated using the following equation: fasting glucose concentration (mmol/L) × fasting insulin concentration (µU/mL)/22.5. Cortisol and adrenocorticotropic hormone concentrations were assessed using an electrochemiluminescence immunoassay (Cobas 8000 e801 analyzer; Roche, Germany). Serotonin concentrations were determined via liquid chromatography with tandem mass spectrometry using an electrochemiluminescence immunoassay (5500 Qtrap; SCIEX, Washington, DC, USA).

### 2.3. Actigraphy

All participants were equipped with a wrist actigraph (ActiGraph wGT3X-BT; ActiGraph LLC, Pensacola, FL, USA). This device was worn on the non-dominant wrist via a wrist strap, and participants were instructed to wear it continuously throughout the day. The device was configured to sample counts per one-minute epoch. Actigraphy data were subsequently analyzed using the ActiLife software 6 (ActiGraph LLC). Over the two-week study period, parameters such as sleep efficiency, total sleep time, time in bed, wakefulness after sleep onset, and the number of nocturnal awakenings were assessed via actigraphy. Participants recorded their sleep habits daily using a sleep diary to enhance uniformity.

### 2.4. Sleep-Related Questionnaire

A self-report questionnaire was employed to assess circadian preference, mood, and sleep-related parameters. Mental health metrics were evaluated using the Patient Health Questionnaire (PHQ-9), which is used to measure depression severity over the preceding two weeks. Comprising 9 items, the PHQ-9 yields a total score of 27, with a score of ≥10 indicative of major depression [23]. The Korean version of the Morningness–Eveningness Questionnaire (MEQ) was used to gauge circadian preference [24]. Consisting of 19 questions, the MEQ prompts participants to consider their “feeling best” rhythms, indicating preferred sleep time and daily performance in various aspects of everyday life. Scores on the MEQ range from 16 to 86, with a higher score reflecting a morning preference.

Overall sleep quality and sleep disturbance were assessed using the Pittsburgh Sleep Quality Index (PSQI), a widely used diagnostic tool for sleep quality [25]. The total PSQI score ranges from 0 to 21, with higher scores indicating poorer sleep quality. The Insomnia Severity Index (ISI) served as a self-report tool for measurement of subjective symptoms, consequences of insomnia, and the degree of concerns and distress [26]. The total ISI score ranges from 0 to 28, with a higher score indicating more severe insomnia [26]. The Stanford Sleepiness Scale (SSS), a self-rating scale used to measure a patient’s subjective evaluation of sleepiness on a seven-point Likert scale [27], and the ESS, consisting of eight questions and a total score ranging from 0 to 24, were employed to assess daytime sleepiness [28]. A higher score on the ESS indicates a higher level of daytime sleepiness.

### 2.5. Assessment of Salivary Melatonin

At the beginning (1st day) and end (14th day) of the intervention, salivary samples were collected to determine the dim light melatonin onset (DLMO). Participants were instructed to avoid eating 1 h before saliva collection and alcohol drinking 12 h before saliva collection. Participants had to chew a specially designed piece of cotton from a container (Salivette, Sarstedt, Germany) with their molars for more than one minute to allow the cotton to absorb enough saliva. After collection, samples were frozen at or below −20 °C within four hours of collection. Saliva samples were collected a total of 10 times per 30 min, starting from 5 h before bedtime up to immediately before sleep. The analysis of saliva melatonin was conducted at an external analytical institution (Global Clinical Central Lab), and the Salimetrics^®^ Melatonin Enzyme Immunoassay Kit (Salimetrics, State College, PA, USA) was utilized for the analysis according to the manufacturer’s protocol. The baseline melatonin concentration was determined as the mean and 2 SDs of three measurements. The point at which the melatonin concentration exceeded this value was suggested as a DLMO in the current study [29].

### 2.6. Real-Time Reverse Transcription (RT) Polymerase Chain Reaction (PCR) Analysis of Clock Genes

Gene expression analysis was performed on peripheral blood mononuclear cells (PBMCs). Blood samples were collected in ethylenediaminetetraacetic acid tubes. PBMCs were isolated via density gradient centrifugation at 3000 rpm for 30 min in Ficoll–Paque medium (GE Healthcare Life Sciences, Pittsburgh, PA, USA). Total RNA was extracted using TRI reagent, and cDNA synthesis was carried out with 1 μg of total RNA using an RT-PCR Kit (Takara Bio Inc., Shiga, Japan). The qPCR primers for CLOCK, BMAL1, PER1, PER2, PER3, CRY1, CRY2, REV-ERBα, and REV-ERBβ were designed. Polymerase chain reaction was carried out using a Thermal Cycler Dice Real-time System (Takara Bio Inc.) and gene-specific primers designed from sequences obtained from the NCBI nucleotide sequence database (Appendix A). PCR was conducted using a Thermal Cycler Dice Real Time System (Takara Bio Inc.) with the following conditions: 40 cycles of denaturation at 95 °C for 10 s, annealing at 60 °C for 10 s, and extension at 72 °C for 10 s. Relative gene expression was determined using the comparative Ct method, and gene expression was calculated using the 2^−ΔΔ^Ct method to ascertain the fold difference between ^Δ^Ct of the target sample and ^Δ^Ct of the calibrator sample. All reactions were performed in triplicate.

### 2.7. Treatment

Participants were randomly assigned in a 1:1 ratio to either the control or LT groups using a centralized, computer-generated system. A total of 40 eligible participants were enrolled in this study, 20 in each group. No dropouts occurred during the two-week period (Figure 1). The LT group was equipped with a lamp manufactured by Fine Technix (Gyeonggi, Republic of Korea). The participants received training to activate the lights using an application installed on their mobile phones before bedtime, allowing the sleep scenario to be initiated. Approximately 15 min before bedtime, participants could initiate the sleep scenario by pressing the sleep scenario start button on the application or directly on the lamp. Once activated, the sleep scenario on the lamp began to operate according to a programmed sequence. The lamp gradually dimmed, in 1 min intervals, from a brightness of ≤30 lux at a color temperature of 2200 K. After 15 min, it automatically turned off as a part of the sequence. Results of the photometer assessments performed by the Korea Photonics Technology Institute are presented in the Appendix A.

### 2.8. Statistical Analysis

Data are expressed as mean ± SD values for continuous variables and as counts (percentages) for categorical variables. Baseline characteristics between the two groups were compared using an independent *t*-test for continuous variables and the chi-squared test for categorical variables, such as smoking status, drinking status, exercise, diabetes, hypertension, and dyslipidemia. Within-group differences after the two-week period were assessed using the paired *t*-test. Changes between groups were analyzed using linear regression, adjusting for each baseline value and BMI. The difference in sleep parameters over the two weeks were analyzed using a linear mixed model with BMI adjustment to consider repeated measure data. The patterns of change in melatonin over the day were identified using restricted cubic spline regression model with advancement in DLMO. Statistical significance was set at a two-sided *p*-value less than 0.05. All statistical analyses were conducted using R software (version 4.1.1; R Foundation for Statistical Computing, Vienna, Austria). The detailed R codes were described in the Appendix A.

## 3. Results

A total of 40 eligible participants (age, 40.0 ± 10.2 years), 20 in each group, were enrolled in this study. Table 1 summarizes the baseline characteristics of the control group (*n* = 20) and LT group (*n* = 20). The demographics, including age range and sex distribution, were similar between the groups. The prevalence of health conditions (diabetes, hypertension, and dyslipidemia) and lifestyle factors (smoking, alcohol consumption, and physical activity) were also comparable. A significant difference was noted in the mean BMI, with the LT group exhibiting a higher BMI (24.8 ± 3.5 kg/m^2^) than the control group (22.19 ± 2.80 kg/m^2^, *p* = 0.013). Additional analyses were performed to mitigate the potential influence of this BMI disparity on the study outcomes.

Table 2 summarizes the changes in biochemical parameters in the two groups before and after the two-week intervention period. No significant differences were observed between the groups. 

Figure 2 demonstrates the changes in metrics related to sleep and mood assessed using questionnaires in both groups after adjusting for each baseline score and BMI. In the control group, no significant changes in the MEQ, ESS, ISI, PHQ-9, PSQI, or SSS scores before and after the two-week period were observed. In the LT group, the ISI, PHQ-9, and PSQI scores significantly improved after the intervention (ISI: 10.25 ± 5.28 to 6.60 ± 4.28, *p* = 0.001; PHQ-9: 6.60 ± 4.33 to 4.80 ± 4.19, *p* = 0.014; PSQI, 10.80 ± 3.86 to 7.60 ± 2.93, *p* = 0.001). No significant changes in the MEQ, ESS, or SSS scores were observed. However, upon comparison between the two groups, only the improvement in the PSQI score was significant in the LT group compared to the control group (difference in difference [95% confidence interval {CI}] = −2.00 [−3.58, −0.43]) after adjusting for baseline PSQI score and BMI. After additionally adjusting for diabetes, hypertension, and dyslipidemia, the significant results remained.

The detailed numbers are presented in the Appendix A.

Table 3 depicts the changes in actigraphy-based sleep parameters within the two groups over the two-week period. Sleep efficiency was significantly higher in the LT group than that in the control group (LT: 84.98 ± 8.47 vs. control: 82.11 ± 12.57, *p* = 0.032). The average number of awakenings were significantly lower in the LT group than that in the control group (LT: 3.27 ± 2.54 vs. control: 3.92 ± 3.71, *p* = 0.048).

As demonstrated in Figure 3, the LT group exhibited an advancement in DLMO of approximately 30 min compared to the control group. This indicates an earlier bedtime and underscores the efficacy of the intervention in modulating circadian rhythms. In contrast, the control group did not display significant changes in DLMO.

Figure 4 presents the results of the post-intervention analysis of circadian clock gene expression in both groups. The LT group exhibited a statistically significant decrease in CRY1 gene expression, with a mean difference of −0.69 (*p* = 0.017), whereas the CLOCK gene’s expression significantly differed between the groups, with a “difference in difference” value of −0.58 (95% CI: −1.09 to −0.07, *p* = 0.027). No other circadian genes exhibited significant changes in expression or differences between the control and LT groups.

### Compliance and Safety

All the participants completed the study (*n* = 40). The safety assessment consisted of an evaluation of adverse events. Although one individual reported experiencing excessive brightness, no serious adverse events were reported.

## 4. Discussion

In our study, the LT parameters were chosen to optimize their therapeutic effects on circadian rhythm regulation and sleep quality improvement. The selection of ≤30 lux intensity, 2200 K color temperature, and a 15 min exposure duration before bedtime was based on their proven efficacy in modulating circadian rhythms and improving sleep quality without impeding sleep onset supported by the seminal work of Figueiro et al. [30] and Chellappa et al. [6], who demonstrated the importance of these parameters in circadian rhythm entrainment and sleep facilitation. Furthermore, the observed reduction in CRY 1 gene expression post-LT in our study provides a molecular insight into how these parameters might influence circadian signals and sleep regulation, providing a basis for their selection and application.

In this study, we investigated the effects of pre-sleep dim LT on sleep-related parameters in adults with sleep disorders, focusing on its potential mechanisms. Short-wavelength blue light exerts a powerful, non-visual impact on behavior and physiological functions, including sleep–wake regulation, cognitive function, and hormone secretion. Most previous studies were focused on the effects of nighttime light exposure on non-visual responses, with light perceived by intrinsically photosensitive retinal ganglion cells being projected to the suprachiasmatic nucleus, significantly influencing melatonin secretion in the pineal gland, an effect that is highly pronounced for short wavelengths of light [31]. In our study, we used dim light and minimizing blue light before bedtime to evaluate not only subjective (self-reported) and objective (actigraphy) sleep outcomes but also classical markers of the circadian system, thereby confirming the impact of LT on the human circadian system. The improvements in subjective sleep-related parameters observed with pre-sleep LT treatment are consistent with previous results 21]. However, artificial dusk lighting studies have not fully addressed the effects of dim LT on circadian rhythm modulation via actigraphy results and classical markers of the circadian system, namely melatonin and clock gene expression. Our study, building upon previous research [21,32], confirmed the positive impact of LT on sleep, encompassing both physiological and molecular outcomes. Our dual approach of objective actigraphy data and a molecular analysis of clock genes provided a more robust understanding of the topic than studies focused solely on subjective measurements and lacking molecular insights.

Several possible explanations for the collective effects of dim LT on sleep-related parameters observed in our study should be discussed. First, our study revealed a tendency for reduced nighttime awakenings when LT that minimizes blue light is used, aligning with the results of a study by Scott et al. [32]. These results suggest that exposure to blue-depleted LED modules in the evening is less disruptive to the circadian system than normal lighting, thereby enhancing overall sleep quality [20,33,34,35]. Second, the advancement of DLMO in our study may be attributed to the specific wavelength and intensity of the warm-colored light used, consistent with the results of a study by Gooley et al. [34], revealing that light with a shorter wavelength more strongly suppresses melatonin than light with longer wavelengths. Our results support the notion that exposure to warm light before bedtime, with reduced blue light, can aid melatonin production and facilitate sleep onset. Notably, our study demonstrates that even 30 min of dim LT before bedtime can enhance DLMO, in line with previous research indicating that exposure to dim light during crucial periods of the circadian cycle can enhance DLMO [36,37,38]. Short exposures of several minutes each or intermittent bright light also influence DLMO, as corroborated by our results showing that a brief exposure before bedtime can facilitate a quick transition to sleep mode in the circadian system. While we did not specifically explore circadian rhythm disorders in this study, our results suggest the potential benefits of LT timing for circadian alignment. Third, investigating clock gene expression was an important aspect of our study. To date, few studies have been conducted to examine the relationship between clock gene expression and pre-sleep dim LT in patients with sleep disorders [8]. We observed a decrease in CRY1 gene expression in the LT group, providing a molecular basis for the potential effect of LT in strengthening the body’s intrinsic circadian signals for sleep. We discovered a significant difference in CLOCK gene expression levels between the LT group and the control group, with the average mRNA level of the CRY1 gene significantly decreasing from 2.27 to 1.59 in the LT group. This pattern may offer a molecular basis for the potential effect of LT in reinforcing circadian signals for sleep. Animal models have revealed that melatonin induces a rhythmic expression of genes such as CLOCK and CRY1 in the pars tuberalis of the pituitary gland [39,40]. Furthermore, the study by Figueiro et al. [30] indicated that exposure to various light spectra can influence circadian phase shifts via the regulation of gene expression. However, additional research is needed to investigate the expression of various clock genes in the peripheral blood under different conditions, including individual variability in clock gene expression and the timing, duration, and intensity of light exposure [40]. The improvements in sleep quality metrics (ISI, PSQI) and PHQ-9 scores observed with pre-sleep LT treatment underscore its potential as a versatile intervention for both insomnia and depression. These promising results underline the possibility of integrating LT into clinical practices for a holistic approach to insomnia treatment, potentially extending to depression management. Given the comorbidity of insomnia and depression, our study advocates for further research to survey how tailored LT interventions could serve as an adjunct or alternative to traditional pharmacological treatments, addressing both sleep and mood disorders concurrently.

In the current study, we did not observe significant changes in metabolic parameters such as blood lipids and peripheral hormone level. It is estimated that longer-term studies will be necessary to elucidate the relationship between light therapy, sleep, and metabolic parameters.

Our study has several limitations. First, the small sample size and short duration of the study limit the generalizability of the results. A short 2-week study duration is inadequate for evaluating the long-term effects of light therapy and may not provide a sufficient timeframe for symptom assessment. Second, the specific demographics of the participant group might have influenced the outcomes. While our study recruited participants voluntarily through recruitment notices, there was a higher proportion of female participants. In future research, we will aim to conduct studies considering a sufficient sample size and a more diverse participant pool, followed over a longer period to validate our results. Third, as the trial period was only two weeks, the study could not determine whether LT is efficacious for the long-term treatment of sleep disorders. Fourth, whether clock gene expression always decreases after LT cannot be proven from our study. Gene expression changes can be influenced by various factors, including light intensity, color temperature, exposure duration, and the characteristics of the target population. Studies to date have not provided consistent results regarding the expression of these genes, necessitating further investigation and verification. Fifth, to accommodate the discomfort of the participants, we only measured saliva up until just before falling asleep. Additionally, saliva melatonin samples were instructed to be collected from each subject’s home environment starting 4–5 h before bedtime, in a dark and quiet setting. However, a limitation exists as subjects did not have complete control over their sleeping environment. Therefore, the DLMO we measured may not be entirely accurate. Consequently, rather than aiming to precisely measure the exact time of DLMO, we focused on confirming a shift of approximately 30 min earlier in the time estimated as DLMO. Sixth, in this exploratory study, we found meaningful changes in the PSQI. The power was 0.958. Based on this pilot study, a corroborative randomized controlled trial with a sufficient sample size is needed. Lastly, we did not perform subgroup analyses on the different types of sleep disorders. Future research should consider individual circadian rhythm variations and types of sleep disorders in patients to explore individualized light exposure times. Despite these limitations, our study is the first in which the effects of LT were investigated on sleep-related parameters, hormone levels, and circadian gene expression. 

## 5. Conclusions

Our study provides substantial evidence supporting the feasibility and efficacy of pre-sleep dim LT in improving sleep quality and efficiency for adults with insomnia. By carefully examining LT parameters and integrating multi-modal assessments, we demonstrate LT’s potential impact at both the physiological and molecular levels. Looking forward, the promising results for managing insomnia and the exploratory findings suggesting benefits for depressive symptoms emphasize the need for larger, more diverse studies. Such further research will be necessary to validate our results, study the nuances in LT’s effects on different populations, and potentially expand its applications within sleep medicine and mental health.

## Figures and Tables

**Figure 1 medicina-60-00632-f001:**
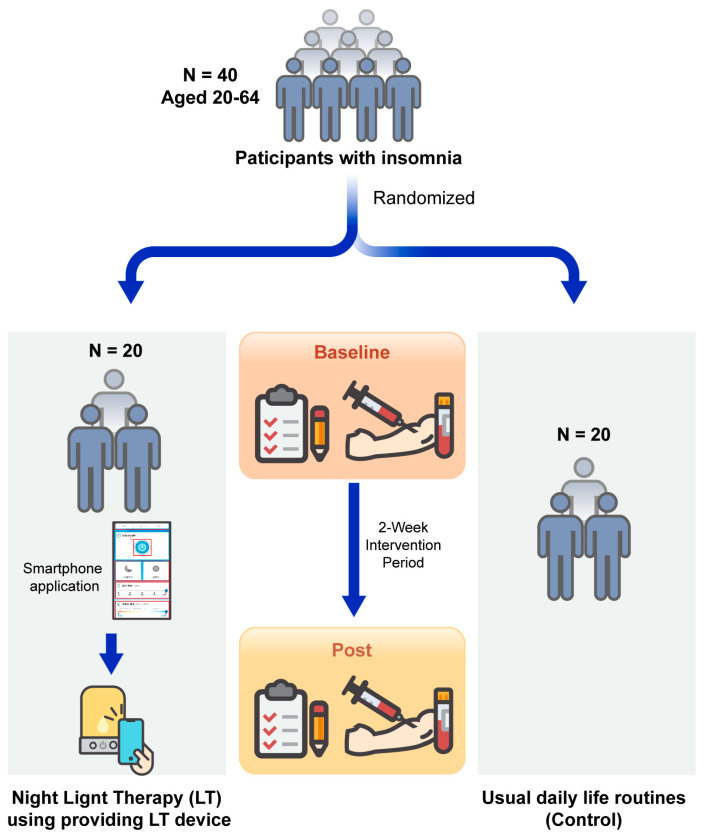
Scheme of study.

**Figure 2 medicina-60-00632-f002:**
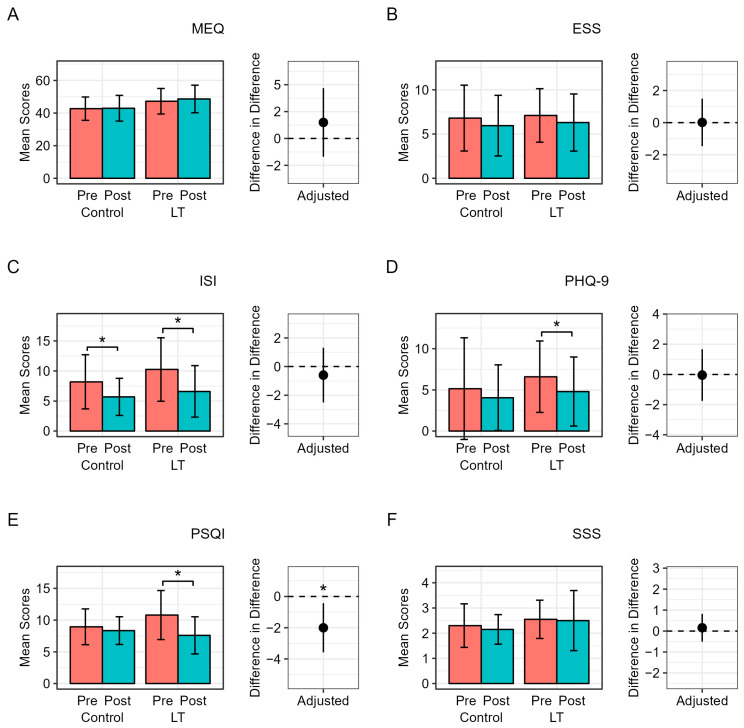
(**A**–**F**) Changes in metrics related to sleep and mood assessed using questionnaires in both groups. Abbreviations: MEQ—Morningness–Eveningness Questionnaire; ESS—Epworth Sleepiness Scale; ISI—Insomnia Severity Index; PHQ-9—Patient Health Questionnaire-9; PSQI—Pittsburgh Sleep Quality Index; SSS—Stanford Sleepiness Scale. The x-axis of the bar graph represents the mean values of each metric related to sleep and mood. The magenta bars represent the pre-intervention stage in both groups, while the teal bars indicate the post-intervention stage. The difference-in-differences and *p*-values were calculated using linear regression to adjust for each baseline value of metrics and body mass index. * *p* < 0.05.

**Figure 3 medicina-60-00632-f003:**
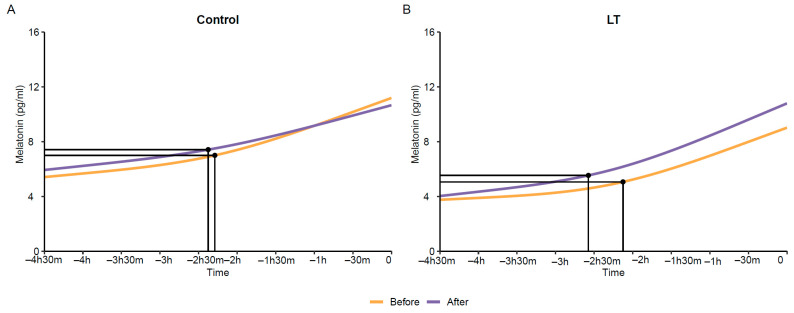
Advancement in dim light melatonin onset (DLMO). (**A**) Control group; (**B**) Light therapy group. The X-axis represents the time intervals, collected in 30 min increments before falling asleep. The Y-axis represents the average value of saliva melatonin. Time (T) 1: 5 h and 30 min before falling asleep; T2: 4 h before falling asleep; T3: 3 h and 30 min before falling asleep; T4: 3 h before falling asleep; T5: 2 h and 30 min before falling asleep; T6: 2 h before falling asleep; T7: 1 h and 30 min before falling asleep; T8: 1 h before falling asleep; T9: 30 min before falling asleep; T10: Just before falling asleep.

**Figure 4 medicina-60-00632-f004:**
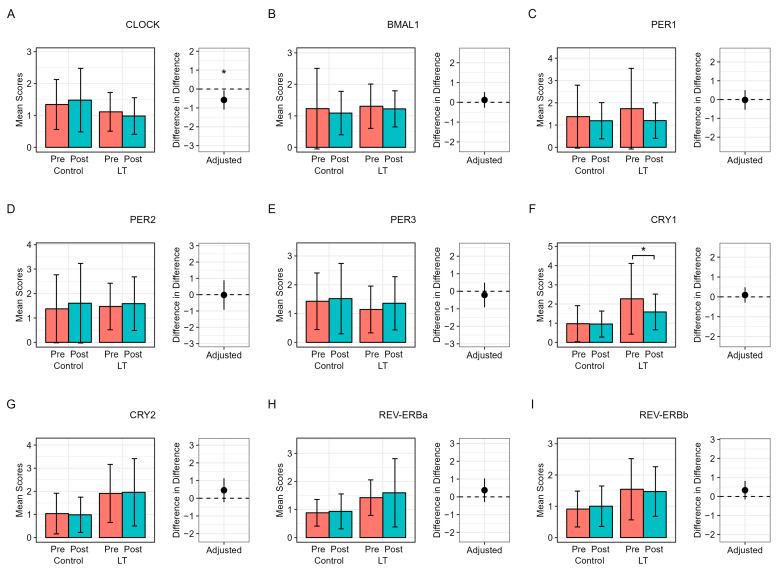
Post-intervention analysis of circadian clock gene expression in both groups (**A**) Expression of CLOCK gene; (**B**) Expression of BMAL1 gene; (**C**) Expression of PER1 gene; (**D**) Expression of PER2 gene; (**E**) Expression of PER3 gene; (**F**) Expression of CRY1 gene; (**G**) Expression of CRY2 gene; (**H**) Expression of REV-ERBα gene; (**I**) REV-ERBβ gene, Abbreviations; CLOCK: Clock Circadian Regulator; BMAL1: Basic Helix-Loop-Helix ARNT Like 1; PER1: Period Circadian Regulator 1; PER2: Period Circadian Regulator 2; PER3: Period Circadian Regulator 3; CRY1: Cryptochrome Circadian Regulator 1; CRY2: Cryptochrome Circadian Regulator 2; REV-ERBα (Nr1d1): Nuclear Receptor Subfamily 1 Group D Member 1; REV-ERBβ (NR1D2): Nuclear Receptor Subfamily 1 Group D Member 2. The x-axis of the bar graph represents the mean values of mRNA levels. The magenta bars represent the pre-intervention stage in both groups, while the teal bars indicate the post-intervention stage. The difference-in-differences and *p*-values were calculated using linear regression to adjust for each baseline value of mRNA levels and body mass index. * *p* < 0.05.

**Table 1 medicina-60-00632-t001:** Baseline characteristics of study sample.

Variable	Overall (*n* = 40)	Light Therapy Group (*n* = 20)	Control Group (*n* = 20)	*p*-Value
Age (year)	40.0 ± 10.2	42.0 ± 9.7	37.9 ± 10.5	0.207
Sex				0.235
Women	32 (80.0%)	18 (90.0%)	14 (70.0%)	
Men	8 (20.0%)	2 (10.0%)	6 (30.0%)	
SBP, mmHg	124.3 ± 14.0	123.2 ± 11.6	125.3 ± 16.3	0.641
DBP, mmHg	75.5 ± 11.1	75.3 ± 10.2	75.8 ± 12.2	0.900
BMI, kg/m^2^	23.5 ± 3.4	24.8 ± 3.5	22.2 ± 2.8	0.013
Smoking, *n* (%)				>0.999
Yes	11 (27.5%)	6 (30.0%)	5 (25.0%)	
No	29 (72.5%)	14 (70.0%)	15 (75.0%)	
Drinking *, *n* (%)				>0.999
Yes	37 (92.5%)	18 (90.0%)	19 (95.0%)	
No	3 (7.5%)	2 (10.0%)	1 (5.0%)	
Exercise †, *n* (%)				0.451
Yes	31 (77.5%)	14 (70.0%)	17 (85.0%)	
No	9 (22.5%)	6 (30.0%)	3 (15.0%)	
DM, *n* (%)				>0.999
Yes	1 (2.5%)	1 (5.0%)	0 (0.0%)	
No	39 (97.5%)	19 (95.0%)	20 (100.0%)	
HTN, *n* (%)				>0.999
Yes	1 (2.5%)	0 (0.0%)	1 (5.0%)	
No	39 (97.5%)	20 (100.0%)	19 (95.0%)	
Dyslipidemia, *n* (%)				0.091
Yes	7 (17.5%)	6 (30.0%)	1 (5.0%)	
No	33 (82.5%)	14 (70.0%)	19 (95.0%)	

Abbreviations: SBP, Systolic blood pressure; DBP, Diastolic blood pressure; BMI, Body mass index; DM, Diabetes mellitus; HTN, Hypertension; *p*-values were calculated via the independent *t*-test. * We defined alcohol consumption as consuming alcohol on one or more occasions per month. † We defined engaging in exercise as participating in moderate-to-vigorous intensity exercise at least three times per week.

**Table 2 medicina-60-00632-t002:** Changes in biochemistry in the two groups before and after the two-week intervention period.

Variable	Light Therapy Group (*n* = 20)				Control Group (*n* = 20)				Difference in Difference (95% CI) ^†^	*p*-Value ^†^
	Pre	Post	Diff	*p*-Value *	Pre	Post	Diff	*p*-Value *		
WBC, 10^3^/μL	6.93 ± 2.2	6.5 ± 1.7	−0.43 ± 1.64	0.252	5.8 ± 1.4	6.2 ± 1.6	0.37 ± 1.37	0.241	−0.27 (−1.29, 0.74)	0.587
Glucose, mg/dL	97.1 ± 9.3	97.2 ± 9.9	0.05 ± 11.59	0.985	92.5 ± 9.5	92.9 ± 9.4	0.35 ± 13.61	0.910	4.66 (−1.31, 10.64)	0.122
Insulin, IU/L	15.76 ± 18.61	15.82 ± 15.19	0.06 ± 20.01	0.989	8.14 ± 3.44	9.90 ± 6.15	1.77 ± 3.84	0.054	4.78 (−2.92, 12.47)	0.216
HOMA−IR	3.97 ± 5.12	4.06 ± 4.53	0.09 ± 5.99	0.946	1.89 ± 0.96	2.26 ± 1.33	0.37 ± 0.79	0.050	1.73 (−0.53, 4.00)	0.130
TC, mg/dL	182.7 ± 33.2	175.6 ± 32.9	−7.05 ± 19.47	0.122	178.9 ± 23.2	181.0 ± 29.8	2.10 ± 19.93	0.643	−13.59 (−28.33, 1.14)	0.069
TG, mg/dL	158.4 ± 132.6	115.0 ± 38.8	−43.35 ± 121.42	0.127	103.0 ± 57.8	102.5 ± 52.6	−0.50 ± 38.58	0.954	−1.95 (−31.69, 27.79)	0.895
HDL−C, mg/dL	60.3 ± 14.3	60.8 ± 12.0	0.50 ± 7.56	0.771	62.8 ± 12.4	64.2 ± 11.4	1.40 ± 6.29	0.332	−1.26 (−5.99, 3.47)	0.590
LDL−C	109.9 ± 33.8	107.4 ± 32.8	−2.50 ± 19.95	0.582	110.6 ± 24.5	113.6 ± 28.3	3.00 ± 18.11	0.468	−11.25 (−24.97, 2.48)	0.105
CRP, mg/dL	1.11 ± 1.38	0.82 ± 0.46	−0.29 ± 1.43	0.368	1.00 ± 1.14	1.98 ± 6.05	0.98 ± 6.13	0.484	−1.71 (−5.03, 1.61)	0.303
Cortisol, µg/dL	9.3 ± 3.6	11.6 ± 5.8	2.4 ± 5.4	0.065	10.2 ± 4.1	11.1 ± 4.0	0.85 ± 4.39	0.395	2.35 (−1.11, 5.81)	0.177
ACTH, pg/mL	23.1 ± 14.8	27.9 ± 26.3	4.7 ± 29.2	0.479	23.8 ± 12.5	27.3 ± 12.1	3.54 ± 13.55	0.257	0.07 (−14.93, 15.07)	0.993
Serotonin, ng/mL	117.0 ± 67.8	123.6 ± 63.6	4.09 ± 47.85	0.746	171.4 ± 68.6	191.0 ± 106.4	19.64 ± 62.25	0.174	−33.05 (−83.63, 17.52)	0.192

Abbreviations: CI, Confidence interval; WBC, White blood cell; HOMA-IR, Homeostatic model assessment of insulin resistance; TC, Total cholesterol; TG, Triglycerides; HDL-C, High-density lipoprotein cholesterol; LDL-C, Low-density lipoprotein cholesterol; CRP, C-reactive protein; ACTH, Adrenocorticotropic hormone. * *p*-values calculated using the paired *t*-test. ^†^ Difference in difference and *p*-value calculated using linear regression to adjust for the baseline value and body mass index.

**Table 3 medicina-60-00632-t003:** Changes in actigraphy-based sleep parameters over two weeks in control and light therapy groups.

Variable	Light Therapy Group	Control Group	Difference (95% CI) ^†^	*p*-Value ^†^
Sleep efficiency	84.98 ± 8.47	82.11 ± 12.57	3.89 (0.52, 7.26)	0.032
Time in bed	443.04 ± 105.79	454.49 ± 139.25	−5.62 (−46.05, 34.82)	0.790
TST	375.01 ± 94.93	368.39 ± 102.27	14.15 (−20.94, 49.25)	0.441
WASO	61.69 ± 44.39	76.91 ± 86.16	−17.94 (−38.12, 2.25)	0.095
Awakenings	19.55 ± 9.46	19.84 ± 11.36	0.26 (−4.31, 4.82)	0.915
Average awakenings	3.27 ± 2.54	3.92 ± 3.71	−0.89 (−1.74, −0.05)	0.048

Abbreviations: CI, confidence interval; TST; WASO. ^†^ Difference and *p*-value were calculated using linear mixed model with BMI adjustment.

## Data Availability

The data are available upon request from the corresponding author.

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
