# Peer review of "Assessing the Feasibility and Efficacy of Pre-Sleep Dim Light Therapy for Adults with Insomnia: A Pilot Study"

_medicina, 2024, doi:10.3390/medicina60040632_

Round 1

Reviewer 1 Report

Comments and Suggestions for Authors

·         i appreciate the multi-modal assessment combining subjective (questionnaires meq, isi, psqi, ess, sss, phq9?) and objective (actigraphy) sleep measures as well as molecular analyses (melatonin assessments, and analysis of circadian clock gene expression) some comments to further improve.

·         small sample size of only 40 participants limits generalizability so i will suggest to call this feasibility study/pilot study.

·         short 2-week study duration, inability to assess longer-term effects mot scales used measure symptoms in overlap period 2-4 weeks it’s a problematic.

·         lack of diversity in study population mostly women is warrant discussion.

·         no subgroup analyses based on insomnia subtypes was performed maybe using ancova? need to adjust for confounding variables like comorbidities, medications, or other sleep habits. in the introduction this need to be also clarified.

·         limited explanation  provided on statistical analysis methods pls update and expand section 2.9 also include codes as supplemental for sas.

·         in selection diagram explain how were participants screened and confirmed to have insomnia disorder versus other sleep disorders. also where there any criteria used to exclude participants with circadian rhythm sleep-wake disorders, which could influence response to light therapy.

·         more discussion need to be regarding light intensity, color temperature, and exposure duration used in the light therapy and their effects.

·         table 1 define drinking and exercise thresholds. e.g. drinking more than 2 units per week etc. caffeine was taken?

·         fig 1 need to be rotated to avoid confusion.

·         fig 3 dlmo need more explanations.

·         who performed acrography reading?

·         was there any adverse effects for light therapy?

·         you have promising pilot results how to put them into clinical applications for insomnia management. can it also used for depression pls discuss based on phq9.

Reviewer 2 Report

Comments and Suggestions for Authors

Thank you for this very interesting study, that has very high interest for both patients and professionals asked to prescribe effective treatments. The paper could be significantly improved with fixing some points.

Line 90: it is not "exclusion criteria" but "non inclusion cirteria", since it is used before including the patients in the study. This is important because "exclusion" means that people who had been previously included, were excluded post-hoc. In such case, you should have an "Intention To Treat" and a "Per Protocol" analysis.

Line 100 giving the number of included patients, must be in Results section.

On line 104 and following ones, should be entitled "Treatment" and not in randomisation and Study Protocol" section. This last title is quite confusional. Randomisation is easier to present and understand in the end of the whole description of the Mat and Meths section, just before the Statistics.

In your calculation for the number of patients, required to have a chance to reach significance, you use data related to the ESS, but this scale will not be used in your list of indicators measured during the study. Please justify it or base your calculation on another scale.

Between your parameters, you mention melatonin on paragraph 2.7. The method to harvest saliva is clearly described, but we don't get any information about the way you measure melatonin. Please provide this information.

Paragraph 2.8: you don't give the list of genes you will study and moreover you don't give any detail about your methodology, the information is too vague.

We don't find details about biochemical analysis in the Mat and Meths, but we find results about it in the Results section. Please give the information.

In the Statistics section, you just mention Student t test, but on reading the results, we find other methods (e.g. homoscedasticity), you must detail and justify more.

The Discussion would be more readable and interesting, if you stratify it, coming from the discussion of your results (including self-criticism of the quality and significance), to emphasize it with comparing it to literature and finish with the Conclusion.

Round 2

Reviewer 1 Report

Comments and Suggestions for Authors

thank you for addressing my concerns 

Author Response

We appreciate the time and effort you have dedicated to providing insightful feedback on ways to strengthen our paper. 

Reviewer 2 Report

Comments and Suggestions for Authors

Thank you for your revisions and modifications. The paper is now much easier to read and is very interesting.

Author Response

(The authors gave the same response as above.)
